# SigScope: Detecting and Understanding Off-Chain Message Signing-related Vulnerabilities in Decentralized Applications

## Abstract

In Web 3.0, an emerging paradigm of building decentralized applications or DApps is off-chain message signing, which has advantages in performance, cost efficiency, and usability compared to conventional transaction-signing schemes. However, message signing burdens DApp developers with extra coding complexity and message designing, leading to new security risks.

This paper presents the first systematic study to uncover and characterize the security issues in off-chain message signing schemes and the DApps built atop them. We present a holistic static-analysis framework, SigScope, that uniquely combines the insights extracted from DApp frontend code (HTML and Javascript) off-chain and backend smart contracts on-chain. We evaluate SigScope using the top 100 DApps to showcase its effectiveness and efficiency. Further, we leverage SigScope to study a large dataset of 4937 real-world DApps and show that 1579 DApps (including 73% of the top 100) rely on the off-chain message signing feature, and 1154 contain vulnerabilities. Finally, we use two real-world vulnerabilities in popular DApps to showcase our findings.

## ACM Reference Format:

Anonymous Author(s). 2018. SigScope: Detecting and Understanding Off-Chain Message Signing-related Vulnerabilities in Decentralized Applications. In *Proceedings of Make sure to enter the correct conference title from your rights confirmation emai (Conference acronym 'XX)*. ACM, New York, NY, USA, 16 pages. https://doi.org/XXXXXXX.XXXXXXX

## 1 Introduction

In Web 3.0, a decentralized application (DApp) provides a web-2.0 compatible user interface (e.g., HTML and Javascript) and runs on a decentralized back-end (i.e., blockchains and smart contracts), which is different from a web-2.0 application running on centralized web servers. By avoiding the centralized trust and associated risks, DApps have gained proven popularity in cryptocurrencies, decentralized finance or DeFi [19, 40, 48, 61], decentralized resource markets [1, 43], decentralized social networks [11], and other domains. For instance, applications in DeFi have experienced a whopping 124% year-over-year (YoY) increase in unique active wallets (UAW) with a total value locked (TVL) worth 103B USD in 2023 [22].

Conventionally, an off-chain DApp user, like a wallet, signs a transaction of a fixed format (e.g., an Ethereum transaction of *from*, *to*, *value*, and *data* fields), and the blockchain network receiving the

transaction verifies the signature in the blockchain-client software (e.g., in Ethereum Geth [9]) before relaying it to the smart-contract layer. However, this workflow incurs transactions per wallet use and is slow, expensive (e.g., long transaction finality and increasing prices or demand over limited supply), and of limited usability (e.g., accessible only to users holding gas fee like Ether). To solve these issues, a modern DApp workflow, known as off-chain message signing [51], is proposed and adopted, which allows the user to sign an application-specific "message" instead of a fixed transaction and lets a DApp web server aggregate and bundle multiple messages into a transaction sent to the blockchain. On the blockchain, signature verification occurs in smart contracts instead of the blockchain clients. Message signing achieves better performance and usability at lower cost (e.g., by transaction bundling [62, 63] and supporting zero-Ether transaction senders [8, 10]), and enjoys a wide and increasing adoption in practice: Message signing has been used as an indispensable primitive in constructing rollups and state channels [7, 12–14], meta transactions [6], account abstraction [8] and many other blockchain-scaling solutions. Our measurement study on DappRadar [4] shows 73% of the top 100 EVM-compatible DApps there depend on message signing.

With the advantageous performance and usability comes extra coding complexity. Message signing, unlike transaction signing, requires DApp developers to design and decide the security-critical function in signing and verification. Unfortunately, designing application-specific signatures is error-prone. Specifically, the DApp developers have to design what information to include in the signed "message", where incomplete message designs can reintroduce classic man-in-the-middle risks to the system, like replay attacks. Besides, the fast-evolving APIs and EIPs (Ethereum Improvement Proposals (EIPs) [2, 3]) for message signing makes it hard for developers to get their API use correct and secure. For instance, there are currently five distinct signing methods implemented, namely *eth_sign*, *personal_sign*, *eth_signTypedData_v1*, *eth_signTypedData_v3*, and *eth_signTypedData_v4*. Yet, it is not well documented in terms of the security implications of these APIs, which caused confusion to even the most skilled DApp developers from top DApps and resulted in API misuse and insecurity, as shown in our research.

**Problem.** This paper tackles the wide open resource problem in the field: *Detect and understand the security issues in off-chain message signing in emerging DApps*. Despite the large body of security research by smart contract analysis (e.g., static analysis [15, 32, 52, 53, 57], symbolic execution [20, 31, 39, 41, 46, 47, 58], fuzzing techniques [17, 30, 34, 45, 64, 65, 67, 70], and applications of formal methods [16, 25, 35, 49, 54, 60]), there is a lack of research on understanding the emerging issues caused by off-chain message signing. Specifically, the existing research focuses on understanding the security of backend smart contracts and often overlooks the frontend components. Given that message signing entails the use of

cryptographic API (e.g., *ecrecover*) in smart contracts, the existing research detecting cryptographic misuses like CRYSOL [70] ignores the insecurity caused by the frontend or the interaction between the frontend and backend, rendering it ill-suited for a comprehensive understanding of DApp security. VetSC [25] presents a study for DApp safety vetting by evaluating both frontend and backend elements. However, its purpose is to derive information from the frontend user interfaces by using natural language processing (NLP) techniques to understand smart contract code semantics. VetSC does not conduct program analysis of the frontend code and is inapplicable to finding the target vulnerabilities of this work, in which a substantial amount of the code related to off-chain message signing is situated within the frontend segment, such as HTML and JavaScript, and analyzing off-chain code is necessary. Overall, understanding the insecurity of off-chain message signing entails the holistic program analysis across a DApp's front and backend.

**Solution.** To address these challenges, we conduct the first systematic study to uncover and understand the security issues in off-chain message signing by holistically assessing its entire workflow on- and off- blockchains. We specifically analyze all message-signing methods available to DApp front-ends and their corresponding verification procedures in the back-end. As a result, we compile a catalog of vulnerability patterns related to off-chain message signing. More importantly, we propose an innovative hybrid analysis technique that automatically identifies pattern-matching vulnerabilities through interprocedural static program analysis applied cohesively to both front-end and back-end code. Specifically, given a DApp, we conduct a pre-processing phase to identify all the signing and verification methods and eliminate the unreachable ones. In the next phase, we employ a proposed algorithm to analyze the back-end smart contract by performing static program analysis to pinpoint the essential security features of off-chain message signing. Our analysis relies on the intrinsic characteristics of these features to identify them. Based on the extracted information, we further analyze the front-end part of the DApp to comprehend the usage of these security features and examine if they are correctly enforced throughout the entire signing and verification process. Finally, we organically combine the analysis results from both the front-end and back-end to infer security vulnerabilities.

We implement and evaluate a prototype SigScope and conduct a study with representative datasets containing 4,937 DApp, including the top 100 DApps from DappRader [23]. The evaluation results show that our tool can effectively and efficiently detect off-chain message signing-related vulnerabilities. Furthermore, our large-scale study discovers a total of 1154 vulnerable DApps in the real world, including some extremely popular ones. Finally, we conduct a case study using real-world vulnerabilities in Compound [19] and Synthetix [56] to showcase the efficacy of SigScope.

**Contributions.** The contributions are summarized as follows:

• We analyze off-chain message signing methods from a security perspective, defining signing-related and verification-related security vulnerabilities and outlining various attack scenarios associated with these security concerns.

• We design and implement a novel automated static analysis framework, SigScope, which performs hybrid analysis on both the front-end and back-end of DApps to detect and analyze off-chain message signing methods and associated security vulnerabilities.

• We evaluate SigScope using a large dataset that includes the top 100 DApps and conduct a large-scale study. The results indicate that 1579 DApps (including 73% of the top 100) rely on off-chain message signing, and 1154 are vulnerable to attacks. We have reported these vulnerabilities to the respective DApp creators.

• We make our prototype implementation and all study data/artifacts publicly available as open-source to facilitate further research[1].

## 2 Background & Off-chain Message Signing

In Appendix A.1, we provide background knowledge on DApps, their interactions with users, and the underlying blockchain. We also describe describe how off-chain message signing works, offering insight into the extremely fragmented signing methods.

## 2.1 Off-chain Message Signing Workflow

Off-chain message signing allows users to sign arbitrary messages off-chain using their private keys and let DApps verify the signed messages subsequently. It is an effective method to avoid significant time and gas costs from traditional on-chain transactions. For instance, for exchange DApps, this approach is widely employed to pre-authorize transactions, allowing the receiver to process signed orders from the sender promptly. Rather than executing buy or sell orders directly on the blockchain, the sender signs messages detailing the orders with his private key. These signed messages are then placed in an off-chain order book. The receiver can quickly access and execute these orders off-chain, bypassing the immediate need for on-chain transactions and avoiding the related gas fees. The validity of these off-chain orders can later be verified by the DApp's back-end smart contract.

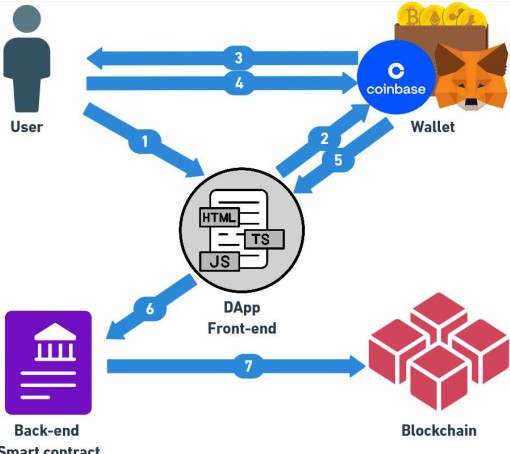

**Figure 1: Off-chain Message Signing Workflow**

The entire workflow of off-chain message signing, depicted in Figure 1, encompasses seven steps and involves five key entities: end users of a DApp, a cryptocurrency wallet, the DApp's front-end, the back-end smart contract, and the underlying blockchain. A cryptocurrency wallet is a tool that enables users to securely store

---

and manage their account keys, execute transaction broadcasts, and facilitate the sending and receiving of cryptocurrencies and tokens. Additionally, it serves as a secure gateway for connecting to DApps. Popular wallets like MetaMask [42] offer JavaScript APIs compliant with EIP-1193 [26] standards. These APIs empower DApps to engage in various activities, such as requesting user accounts, accessing data from connected blockchains, prompting users to sign messages and transactions, and more.

The off-chain message signing process begins when the user engages with the DApp's UI (e.g., HTML) within the front-end. The user inputs data and initiates actions, such as executing a token transfer (Step 1). Following this, the front-end of the DApp presents its requests, such as requesting message signing for a token transfer, to the user via his/her cryptocurrency wallet, utilizing the Wallet's JavaScript APIs (Steps 2 & 3). Subsequently, the user can react to these requests by either approving or rejecting them, conveying their decision to the DApp's front-end through the wallet (Steps 4 & 5). Upon approval, a signed off-chain message from the user is created and relayed to the DApp. The DApp then forwards this signed message to its back-end for verification and processing, invoking relevant functions within the smart contract (Step 6). If the verification succeeds and the DApp's call is valid, it can modify the blockchain state, such as transferring tokens (Step 7).

Throughout the procedure, cryptographic signatures are fundamental in authenticating the source and ensuring the consistency of messages within off-chain computation contexts in DApps. Serving as a foundational element in blockchain technology, these signatures, as a primitive, underpin various aspects of blockchain. They confirm that the signer has consented to a specific message or transaction, thereby offering verification to the smart contract.

## 2.2 Signing Methods

Each blockchain account comes with both a public and a private key. With the private key, accounts can sign data off-chain, creating a unique signature for that specific data. Cryptocurrency wallets and assorted libraries present various signing methods, as detailed in Table 1, based on multiple EIPs [2, 3]. These methods are designed to streamline the communication between DApps and users. Nonetheless, they can be a source of confusion and security issues.

**Table 1: Summary of Signing Methods**

| Signing Method | Human Readability | Ease of Impl | Secure Prefix | DS | Data Structure |
|---|---|---|---|---|---|
| eth_sign | No | Easy | No | No | Arbitrary(hash) |
| personal_sign | Yes | Easy | Yes | No | UTF-8 string |
| signTypedData_V1 | Yes | Moderate | Yes | No | Array |
| signTypedData_V3 | Yes | Hard | Yes | Yes | Struct |
| signTypedData_V4 | Yes | Hard | Yes | Yes | Array, Struct, Recursive |

**eth_sign.** This method offers a flexible signing mechanism that can authenticate any hash value, enabling the signing of transactions and diverse data forms. However, *eth_sign* lacks human readability and essential security features, rendering it impractical for actual applications. Notably, *web3.eth.sign* from the *web3.js* library offers a comparable variant of this signing method. This method has been deprecated and should not be used in any DApp.

**personal_sign.** This method operates similarly to *eth_sign* but enhances security by making messages human-readable and further adding a safeguarded prefix, i.e., "\x19Ethereum Signed Message:\n <length of message>", to the message before its hashing and signing. This addition, while bolstering security, also leads to increased

computational load and a higher demand for storage resources compared to *eth_sign*. The functions *web3.eth.personal.sign* from the *web3.js* library and *signMessage()* from the *ethers.js* library provide equivalent capabilities to this method.

**eth_signTypedData family.** This family of methods is associated with the Ethereum Typed Data Signing Standard, denoted as EIP-712 [3]. The primary function entails signing structured data following a predefined schema. These structures often encompass domain-specific messages or authentication requests. One of the hallmark advantages of these methods is their ability to generate highly human-readable signatures while maintaining efficiency in on-chain processing.

There exist three different versions of signing methods in this family. In *eth_signTypedData_v1*, users are limited to signing arrays of primitive fields without any domain separator. *eth_signTypedData _v3* expands functionality by enabling the signing of structs, albeit without support for arrays and recursive data structures, but it has a `Domain Separator`, a.k.a, DS, which consists of four major elements that specify relevant DApp information in the signature's message to prevent signature missuses, including its name, version, blockchain network (referred to as `chainId`), and the designated smart contract responsible for the signature verification process (`Verifying Contract`, a.k.a, VC).

*eth_signTypedData_v4* (Listing 4, Ln.15) also contains a DS (Ln.4) and further introduces the capability to sign arrays, allowing for the signing of structs containing any Solidity primitive field, including arrays and arrays of structs. As of today, it is the most recommended signing method for DApps to interact with users.

## 2.3 Signed Message Verification

After obtaining the signed message, the next phase is to verify the message's authenticity on the blockchain. This requires the DApp to call the relevant contract by submitting the message via a transaction, incurring an on-chain action and gas fee.. Mainstream blockchains (e.g., Bitcoin [44] and Ethereum [27]) leverage Elliptic Curve Digital Signature Algorithm (ECDSA) [5], with signatures comprising three parameters: $r$, $s$, and $v$.

As illustrated in Appendix A.2 Listing 5, Solidity includes a universal accessible function named *ecrecover* (Ln.10). This function, when given the four specific parameters, returns an address. The signature is considered authentic if this address aligns with the signer's address. Once the signature is verified as valid, the final phase involves processing the signed message. For example, in a popular DApp *UniswapV2*, if the signature is successfully authenticated within the *permit* function (Ln.12), it activates the *_approve* function (Ln.16). This function authorizes the DApp to handle the transfer of the token amount specified by the user.

## 3 Off-chain Message Signing Security Analysis

Although off-chain message signing has become extremely popular, few developers know how to use the feature securely. According to our research, even the most skilled developers from well-known DApps are prone to errors or may use deprecated signing APIs like *eth_sign*, resulting in huge security loopholes (more details in Section 5). These security issues can be largely attributed to the intricate nature of the off-chain message signing process. More seriously, to date, there is no comprehensive research into this critical and prevalent technique from a security standpoint.

In this section, we delve into an in-depth examination of the security aspects of off-chain message signing, covering both the signing and the verification stages. We aim to pinpoint the distinctions between various signing methods and scrutinize the verification procedures to uncover potential security vulnerabilities. We then outline attack strategies that could potentially exploit these identified security risks. The research outcome can provide solid insight for DApp developers to use the off-chain message signing securely.

## 3.1 Threat Model

Our foundational trusted elements include (a) blockchains, (b) the runtime environment for smart contracts, (c) web servers delivering DApp services, (d) client-side wallets and browsers, and (e) the communication channels between browsers and services. This work does not address attacks targeting vulnerabilities in the consensus mechanism, peer-to-peer network, or mining. We do not consider lower-level software attacks that compromise the operating system, runtimes, or browsers. Additionally, network-level attacks aimed at interrupting or intercepting traffic are also not within the scope of this work. Our analysis studies the entire off-chain message signing flow, including both signing and verification. We focus on application-level vulnerabilities pertaining to the implementation of smart contracts, the design of front-end interfaces, and their interactions within the off-chain message signing process. Firstly, DApp web interfaces may appear harmless but lack completeness or accuracy. Secondly, errors made by developers can result in incorrect or insecure smart contract code, which further affects the security of off-chain message signing. Third, there could be a disparity between the front-end implementation and the internal logic within corresponding smart contract functions. Attackers can exploit these to misuse smart contracts, leading to financial losses.

## 3.2 Formal Definition

First, our study and assessment begin by dissecting every element within the off-chain message signing process and enumerating all the necessary conditions for producing a signature based on defined steps in Figure 1.

**Definition 1.** An off-chain message signing process consists of the following components:

1.**Secure Message Construction:** At the end of Step 1, given a message to sign, DApp's front-end constructs a secure form of the message, referred to as $M$, to mitigate the risk associated with signing a wide range of arbitrary message content, including predefined transactions. This transformation can be achieved through methods such as padding.

2.**Signing Method Selection:** An appropriate off-chain signing method must be selected to sign the constructed secure message M (Step 2 & 3). It is essential to avoid insecure or deprecated APIs during this step and to utilize all the security measures correctly. Then, the user can sign the provided message with his/her own private key, as shown in Equation 1, and send it back to the DApp (Step 4 & 5).

$$Signature = Sign(M, singingAPI, pri\_key) \qquad (1)$$

The next phase verifies signed messages within the back-end verifying contracts (VC). We assess all components in the smart contract verification function.

**Definition 2.** An off-chain signed message verification process within VC consists of the following components:

1. **Signature Validation:** A signed message *Signature* to be verified possess the original message $M$ and the owner's address (Step 6). It is vital to consider the usability of the signature by two properties: 1) the potential number of signature usages and 2) the time window of the signature validity.

2. **Signer Verification:** The VC logic must recover the signer's address from the provided *Signature* and message $M$ by leveraging *ecrecover* method (Equation 2), compare the recovered signer address with the provided owner address (Equation 3), and consider the aforementioned two properties. If all are successfully verified, VC confirms the validity of the *Signature* and moves on to take the requested action (Step 7).

$$Signer_{Addr} = ecrecover(Signature, M) \qquad (2)$$

$$Signer_{Addr} \overset{?}{=} Owner_{Addr} \qquad (3)$$

## 3.3 Signing-related Vulnerabilities

To ensure thoroughness, we meticulously scrutinize each element in the definition of the signing process to identify a comprehensive set of vulnerabilities related to signing.

**Missing Secure Prefix.** A secure prefix of a message, such as "\x19Ethereum Signed Message:\n<length of message>" in *personal_sign*, is crucial to the secure message construction of the off-chain message signing process. This simple yet powerful feature safeguards against the misuse of signatures, where attackers might sign arbitrary data, such as transactions, and employ the signature to impersonate the victim. By prefixing a message, the resulting signature becomes distinctly identifiable as one associated with Ethereum, enhancing its security. Nonetheless, this crucial security measure is not incorporated in *eth_sign*, rendering it vulnerable to attacks. Because of this, *eth_sign* has been disabled by default. However, our study shows that even some top DApps still use this method. It is also noteworthy that this feature merely narrows down the signature's applicability to the Ethereum blockchain and other Ethereum Virtual Machine (EVM)-compatible blockchains. Given that these blockchains host millions of smart contracts, the effectiveness of this security measure is limited.

**Missing or Improper DS.** Domain Separator (DS) is a security feature that limits signatures' applicability to specific DApps. Specifically, a DS includes several fields describing the signing domain, such as the DApp's name, version, chainId that represents the specific blockchain, and VC indicating the address of the verifier contract. A proper DS ensures that a signature made in one domain cannot be misused in another, providing a clear scope of validity for the signed data.

Accordingly, signing methods that do not support this feature (e.g., *eth_sign*, *personal_sign* and *eth_signTypedData_v1*) are inevitably vulnerable to signature replay attacks. Moreover, while this feature is supported by *eth_signTypedData_v3* and *eth_signTypedData_v4*, we find that simply providing the DS structure is not enough to keep DApps from security attacks. Developers must implement it properly in a secure fashion. However, as demonstrated by Listing 1, a popular DeFi Carbon [21] (>20K monthly active users) implements an insecure DS with the incorrect VC and the absence of chainId.

```
1  export const DEFAULT_CARBON_DOMAIN_FIELDS = {
2      name: 'Carbon', version: '1.0.0',
3      verifyingContract: 'cosmos', salt: '1',}
```

**Listing 1: Carbon DApp Domain Seperator**

## 3.4 Verification-related Vulnerabilities

To ensure comprehensive coverage, we meticulously analyze every element in the signature verification process to identify a comprehensive set of vulnerabilities.

**Missing or Improper Nonce.** The first feature, nonce, plays a critical role as it limits the potential number of signature usages for verification by VC, thereby satisfying the first property of the signature validation process. To achieve this, DApp developers must implement nonce correctly in both the front-end and the back-end, as shown in Listing 5 and 6. Any mistake on either side will result in a nonce-related vulnerability.

Particularly, in smart contracts, developers are required to properly configure nonce within a mapping-like data structure, establishing a connection between unique numbers and specific addresses (Listing 5, Ln.1). The nonce should be included within the message content and be incremented following each verification of a signature (Listing 5, Ln.7). On the front-end side, to ensure a signature can be verified, developers must ensure that the nonce value on the front-end aligns with its counterpart on the back-end (Listing 6, Ln.1-6) and incorporate nonce within the body of the message (Listing 6, Ln.14).

**Missing or Improper Deadline.** The next security feature is `deadline`, which defines the time window of the signature validity, limiting its validity to a determined period. This satisfies the second property of the signature validation process. Similar to nonce, it must be implemented correctly on both sides to avoid a deadline-related vulnerability.

First, `deadline` should be defined as the current time (message construction time) plus a certain validity period to allow certain processing and verification overhead but filter out any further signature reuse. In the front-end, developers need to retrieve the current time by extracting the last *Block Number* or *Timestamp* associated with the blockchain and then add some predefined validity period (Listing 6, Ln.8). Then, `deadline` must be part of the message body in both sides (Listing 5 Ln.8 & Listing 6 Ln.15). Lastly, a validating step in the verification process (Listing 5, Ln.4) is needed to check that the signature submission time is not outside *deadline*.

**Missing or Improper Validity Check.** The last security feature is the `Signature Validity Check` (SVcheck). It resides in the signer verification function in smart contracts and checks the validity of the signer address by verifying the recovered address value using ecrecover. To correctly implement SVcheck, the recovered address should be a non-zero value that is equal to the signer's address (Listing 5, Ln.12-14). Without such a correctly implemented validity check, the DApp will contain a validity check-related vulnerability, which could result in a change in the state of the blockchain to the profit of an unauthorized signer.

## 3.5 Proposed Attacks

Based on the aforementioned vulnerabilities, we propose various attack scenarios that can exploit these vulnerabilities, which are detailed in Appendix A.3. To demonstrate, We have successfully implemented these proposed attack scenarios by deploying the front-end of real-world vulnerable DApps ([19], [56], etc) in a local environment and deploying the back-end VC in a testnet using Remix [50] for testing the attacks, along with a *proof of concept* (PoC) to validate their effectiveness.

## 4 SigScope Design and Implementation

To effectively detect off-chain message signing-related security vulnerabilities, we propose SigScope, an automated static analysis framework that performs hybrid code analysis on both the front-end and back-end.

### 4.1 System Overview

Figure 2 presents SigScope's system overview. It takes a DApp as input and generates a security report. The system comprises four main phases: pre-processing, back-end code analysis, front-end code analysis, and security inference.

Pre-processing can be split into two tasks: eliminating unreachable code and identifying signing methods. This phase (detailed in Appendix A.5) focuses on pinpointing all instances of signing and verification methods within the DApp and then sifting out those considered unreachable. Following this, our system undertakes interprocedural static analysis in the DApp's back-end - its smart contracts — to gather essential information about the verification process. Subsequently, SigScope advances to analyze the DApp's front-end, which usually involves HTML and JavaScript (or TypeScript) code, aiming to extract further details pertinent to the message signing procedure. Leveraging the insights acquired from analyzing both the front-end and back-end code, our system ultimately performs a security inference for the DApp, culminating in a comprehensive security report highlighting potential security vulnerabilities related to off-chain message signing. The following subsections describe each phase in depth. Additionally, more implementation details of SigScope are provided in the Appendix A.6.

---

**Algorithm 1** Back-end Code Analysis

---

**procedure** VerificationAnalysis($sc, vf, rc$)
1:   $cg \leftarrow$ CallGraph($sc$)
2:   $F_{cp} \leftarrow$ AllFuncs($cg, vf, rc$)
3:   $ret_{ecrecover} \leftarrow$ ReturnValue($ecrecover$)
4:   $\mathbb{SINK} \leftarrow$ ForwardDataFlowAnalysis($sc, ret_{ecrecover}$)
5:   **if** $\exists sink \in \mathbb{SINK}$ && $sink$ is ConditionCheck **then**
6:    $cmpVal \leftarrow$ ConditionAnalysis($sink, ret_{ecrecover}$)
7:    $\mathbb{SRC} \leftarrow$ BackwardDataFlowAnalysis($sc, cmpVal$)
8:    **if** $\exists src \in \mathbb{SRC}$ && $src$ is Parameter($rc$) **then**
9:     $SVCheck \leftarrow$ TRUE
10:   $\mathbb{ARGS} \leftarrow$ FindAllAguments($F_{cp}$, '$Keccak$')
11:   **for** $\forall arg \in \mathbb{ARGS}$ **do**
12:    $\mathbb{SRC} \leftarrow$ BackwardDataFlowAnalysis($sc, arg$)
13:    **if** $\exists src \in \mathbb{SRC}$ && $src$ is ConditionCheck **then**
14:     $cmpVal \leftarrow$ ConditionAnalysis($src, arg$)
15:     **if** $cmpVal$ is $CurrentTime$ **then**
16:      $POS_{ddl} \leftarrow$ Position($arg$)
17:      $Deadline \leftarrow$ TRUE
18:    **if** $\exists src \in \mathbb{SRC}$ &&
      $src$ dep Parameter($rc$) &&
      $src$ is Incremented &&
      $src$ is Non-Local Mapping Variable **then**
19:     $POS_{nonce} \leftarrow$ Position($arg$)
20:     $Nonce \leftarrow$ TRUE
21: **return** $SVCheck, POS_{ddl}, Deadline, POS_{nonce}, Nonce$
**end procedure**

---

### 4.2 Back-end Code Analysis

After the pre-processing, SigScope performs the back-end code analysis on the smart contracts to extract verification-related information from the verification process of a DApp. This phase is

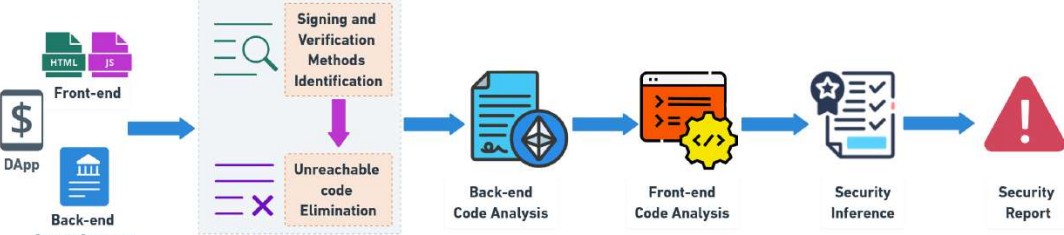

**Figure 2: SigScope System Overview**

responsible for analyzing three major security features - SVCheck, deadline, and nonce. Additionally, it infers the structure of the signed message, which is vital for front-end analysis, specifically the positions of nonce and deadline in the signed message. This is why we perform the back-end analysis first. The biggest challenge of this back-end analysis is identifying nonce and deadline from a message and further recording their positions. A simple approach is to create a keyword list for each element and rely on program symbols, i.e., function and variable names, to identify them. For instance, if a variable in a verification method is named 'deadline', it is probably the deadline in the message. However, this approach will inevitably lead to high inaccuracy since the keyword list can never be exhaustive. Alternatively, we rely on the intrinsic characteristics of these security features to identify them.

Algorithm 1 delineates our detailed algorithm for performing the back-end verification analysis. Essentially, the algorithm takes three inputs: the back-end smart contract ($sc$), the verification function identified in pre-processing ($vf$), and the root caller of the verification function ($rc$). The algorithm begins by generating the call graph of $sc$ and producing $F_{cp}$, which is a set of functions that are covered in the call path from $rc$ to $vf$ in $sc$ (Ln.1-2). Then, the algorithm detects the existence of $SVCheck$ (Ln.3-9). It first identifies the return value of an *ecrecover* function $ret_{ecrecover}$ and performs forward dataflow analysis on it to create a set of statements $\mathbb{SINK}$ (Ln.3-4). If a conditional check $sink$ exists in the set, we analyze the condition of $sink$ to extract the variable that is compared against $ret_{ecrecover}$ (Ln.5-6). We further perform backward dataflow analysis on the variable to extract its data origin. If the data originated from a parameter $rc$, which means the recovered signer of the message is compared against part of the signed message (the original signer), then we believe $SVCheck$ is enforced (Ln.7-9).

Subsequently, the algorithm starts to analyze $Deadline$ and $Nonce$ (Ln.10-20). It first finds all the arguments ($\mathbb{ARGS}$) of the hash function $Keccak$ and performs backward dataflow analysis on each argument $arg$ to generate a set $\mathbb{SRC}$ (Ln.10-12). If a conditional check $src$ exists in the set that compares $arg$ against the current time (e.g., $Blocknumber$ or $Timestamp$), we believe $Deadline$ exists and further record the position of $Deadline$, which is the position of $arg$ in the parameter list (Ln.13-17). Further, if there exists another $src$ in $\mathbb{SRC}$ that is data-dependent on a parameter of $rc$ (the nonce is signer-specific) and is incremented for every verification, and its value comes from a non-local mapping variable, we believe $Nonce$ exists and record its position. Eventually, the algorithm returns the existence of the three security features and the positions of nonce and deadline in the signed message.

---

**Algorithm 2** Front-end Code Analysis

---

**procedure** SigningProcessAnalysis(js, sm, $POS_{ddl}$, $POS_{nonce}$)
1:   $V_{data} \leftarrow$ FindDataParameter(sm)
2:   $\mathbb{SRC} \leftarrow$ BackwardDataFlowAnalysis(js, $V_{data}$)
3:   $message \leftarrow$ FindMessage($\mathbb{SRC}$)
4:   $DS \leftarrow$ FindDS($\mathbb{SRC}$)
5:   $Result_{DS} \leftarrow$ AnalyzeDS($DS$)
6:   **if** $POS_{ddl}$ != $\varnothing$ **then**
7:     $V_{ddl} \leftarrow$ LocateDDL($message$, $POS_{ddl}$)
8:     $\mathbb{SRC} \leftarrow$ BackwardDataFlowAnalysis(js, $V_{ddl}$)
9:     **if** $\exists src \in \mathbb{SRC}$ && $src$ is CurrentTime + period **then**
10:      $Result_{deadline} \leftarrow$ TRUE
11:   **if** $POS_{nonce}$ != $\varnothing$ **then**
12:     $V_{nonce} \leftarrow$ LocateNonce($message$, $POS_{nonce}$)
13:     $\mathbb{SRC} \leftarrow$ BackwardDataFlowAnalysis(js, $V_{nonce}$)
14:     **if** $\exists src \in \mathbb{SRC}$ && $src$ is from back-end **then**
15:      $Result_{nonce} \leftarrow$ TRUE
16:   **return** $Result_{DS}, Result_{deadline}, Result_{nonce}$
**end procedure**

---

## 4.3 Front-end Code Analysis

After analyzing the verification process in the back-end smart contract, SigScope needs to extract and verify the proper implementation of security-related features in the front-end, namely $DS$, $Deadline$, $Nonce$, and Secure Prefix. To this end, SigScope utilizes algorithm 2 to realize this part. The algorithm takes the JavaScript code ($js$), the identified signing methods from pre-processing ($sm$), and the positions of deadline and nonce from the previous phase ($POS_{ddl}$ and $POS_{nonce}$) and outputs the analysis results from $DS$, $deadline$ and $nonce$. Initially, it finds the message data section variable $V_{data}$ by checking the corresponding parameter (e.g., the second parameter in $signTypedData\_v4$) from the signing method $sm$ (Ln.1). The algorithm performs backward dataflow analysis on $V_{data}$ to find the message section $message$ and the domain separator $DS$ since $V_{data}$ is data dependent on both of them (Ln.2-4). We can then analyze $DS$ to see if it enforces valid chainID and verifyingContract (Ln.5). Moving on, the algorithm performs analysis on $deadline$ (Ln.6-10). We first utilize $POS_{ddl}$ from the back-end to locate the variable that contains deadline information ($V_{ddl}$) and perform backward dataflow analysis on it to check whether the value comes from the current time plus a certain small period. If so, we believe $deadline$ is correctly implemented (Ln.7-10). We conduct a similar analysis on $nonce$ to see if its value comes from a call to the back-end and determine the correctness of $nonce$ (Ln.11-15). Eventually, the algorithm returns the analysis results on $DS$, $deadline$ and $nonce$. SigScope also analyzes to detect the existence of Secure Prefix by utilizing pre-processing information about the type of signing methods ($sm$) used. It further analyzes

the front-end to identify patterns such as "\x19Ethereum Signed Message:\n<length>" by parsing the message. This ensures the detection and handling of Security Prefixes in the signing process.

### 4.4 Security Inference

After analyzing both the front-end and back-end, SigScope possesses information regarding all utilized signing methods and the implementations of security features. In the final phase, SigScope consolidates this information to infer all security vulnerabilities related to each signing method based on the defined features and scenarios outlined as signing-related and verification-related vulnerabilities in Section 3. In other words, the analysis report produced by SigScope lists all used signing and verification methods and their associated security vulnerabilities in a DApp.

## 5 Evaluation

In this section, we first evaluate the effectiveness and efficiency of SigScope, then leverage our system to conduct a large-scale study to reveal the real-world impact of off-chain message signing-related vulnerabilities. Furthermore, we conduct a case study to demonstrate the usefulness of SigScope using vulnerabilities in two popular DApps.

### 5.1 Experimental Setup

**Two Datasets.** To evaluate SigScope, we use DApp source code as ground truth and compile two datasets. The first dataset (D1) consists of the top 100 EVM-compatible DApps from DappRadar [23], covering the most popular applications. The second dataset (D2) includes 4,837 real-world DApps from GitHub, selected based on recent updates within the past year and a minimum number of stars and forks to reflect community interest and active development. This approach ensures the datasets represent both widely used and actively maintained DApps.

### 5.2 Effectiveness and Efficiency

We use a dataset containing randomly selected 10% of each dataset (494 DApps in total) and manually extract the ground truth. We execute SigScope on these samples and manually check the results to evaluate SigScope's effectiveness by calculating the F-1 score. Our manual inspection reveals that 159 out of 494 DApps use off-chain message signing methods, and 118 of these DApps contain various vulnerabilities, as defined in Section 3. SigScope successfully detects all 159 DApps using off-chain message signing methods and provides detailed implementation analysis for each. Additionally, it identifies 123 associated vulnerabilities, including issues such as the improper implementation of nonce, with only 5 cases being mislabeled (detailed in Appendix A.8). This results in a 0% false negative (FN) rate and a 4.2% false positive (FP) rate. Consequently, SigScope achieves an impressive F1 score of 97.9%. We also conclude that SigScope's efficiency is sufficient for performing large-scale DApp analysis, as detailed in Appendix A.9.

### 5.3 Real-world Study

Using SigScope, we conduct a large-scale study on the prevalence and security of off-chain message signing in real-world DApps. We analyze both datasets, D1 and D2, totaling 4,937 DApps. Table 2 provides detailed insights into the types and usage of these methods. For D1, 73% of the top 100 DApps employ various off-chain message signing methods, emphasizing their significance. These 73 DApps collectively hold over $17B in assets and involve more than 976,630 unique active wallets (UAW). In dataset D2, SigScope identifies

1,506 DApps using off-chain message signing methods. Across both datasets, we find 1,886 unique invocations of signing methods in 1,579 DApps, as some employ multiple signing methods. This indicates that 32% (1,579 out of 4,937) of DApps actively use off-chain message signing. (Growth trend is provided in Appendix A.10)

**Table 2: Usage of Off-chain Message Signing**

| Signing Method | D1 Count | D2 Count | D2 Percent |
|---|---|---|---|
| eth_sign | 33 | 804 | 53.3% |
| personal_sign | 29 | 522 | 34.6% |
| signTypedData_V1 | 6 | 39 | 2.6% |
| signTypedData_V3 | 9 | 21 | 1.4% |
| signTypedData_V4 | 62 | 361 | 24% |

### 5.4 Off-chain Message Signing Vulnerabilities

After detecting these 1886 unique calls to signing methods in a total of 1579 unique DApps, SigScope extracts and reports security vulnerabilities based on the detailed algorithm mentioned in Section 4 by considering five different security features and their respective vulnerabilities defined in section 3. Table 3 illustrates the missing or improperly implemented signing-related vulnerabilities for each specific off-chain message signing method category.

**Table 3: Categorized Signing-related Vulnerabilities**

| Signing Method | Secure Prefix Missing | DS Missing | DS Improper VC | DS Improper chainId |
|---|---|---|---|---|
| eth_sign | 837 | N/A | N/A | N/A |
| personal_sign | 0 | 551 | N/A | N/A |
| signTypedData_V1 | 0 | 45 | N/A | N/A |
| signTypedData_V3 | 0 | 0 | 2 | 1 |
| signTypedData_V4 | 0 | 0 | 9 | 3 |
| Total | 837 | 596 | 11 | 4 |

SigScope identifies 1,203 unique vulnerable DApps. Upon manual inspection, we confirm that, aside from 49 false positives (detailed in Appendix A.12), SigScope accurately flags the vulnerabilities. This leads to the striking conclusion that approximately 73% (1,154 out of 1,579) of the detected DApps contain one or more vulnerabilities related to off-chain message signing methods, as illustrated in Figure 3a and Figure 3b.

After discussing the number of discovered vulnerabilities in DApps using the off-chain message signing method, it is crucial to consider their impact on the performance and security of DApps within the blockchain ecosystem. Based on the attack scenarios defined in Section 3.5, we further map the presence of off-chain message signing vulnerabilities to the corresponding attacks that can exploit them. Figure 4 presents statistics illustrating the likelihood of attacks on DApps, highlighting the potential impact of these vulnerabilities. To demonstrate, we successfully implement each attack scenario at least twice by deploying the front-end of more than ten real-world vulnerable DApps (e.g., [19], [56]) in a local environment and deploying the back-end VC on a testnet using Remix [50] for testing the attacks. Additionally, we provided a *proof of concept* (PoC) to validate their effectiveness. As a result of the attacks, a single off-chain signed message, originally intended for executing only one specific on-chain transaction, can be abused to alter the transaction's content or to execute the transaction multiple times. For instance, if a DApp is vulnerable, an attacker could use a single off-chain signed message to vote or transfer funds and tokens multiple times, rather than just once.

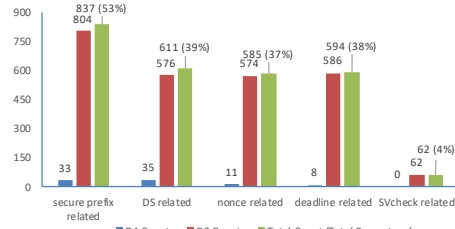

**(a) DApps Vulnerabilities Count**

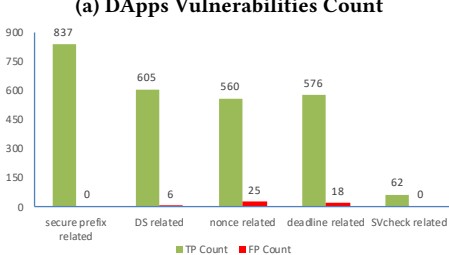

**(b) True Positives & False Positives**

**Figure 3: Study on DApps Vulnerabilities**

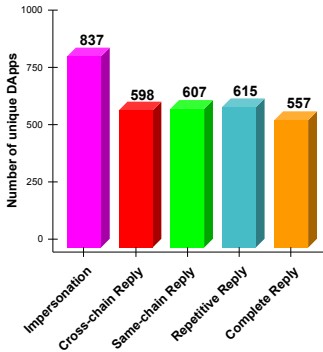

**Figure 4: Possible Attacks Scenarios**

## 5.5 Case Studies

We demonstrate SIGSCOPE's efficacy through two case studies. The first reveals missing security components in a signing method, while the second, detailed in the Appendix A.13, highlights incorrect verification process in a DApp.

**Case 1: Missing security measures.** First, we investigate *Compound*, an extremely popular DApp with a Total Value Locked (TVL) value of $1B+. Listing 2 displays the front-end code of *Compound*, where SIGSCOPE detects that the message consists of only two components: *proposalId* and *support*. While our system is able to find the presence of domain (i.e., the DS), our analysis result shows that there is no nonce or deadline being enforced.

```
1  const message: VoteSignatureMessage = { proposalId
     , support };
2  const types: VoteTypes = {
3    EIP712Domain: [{name:'name', type: 'string'},
4    {name:'chainId', type: 'uint256'},
5    {name:'verifyingContract',type:'address'},]
6    Ballot: [{name: 'proposalId', type: 'uint256'},
7    {name: 'support', type: 'uint8' }]};...
8  const signature = await sign(domain, primaryType,
     message, types, signer); ...}
```

**Listing 2: Compound DApp Front-end**

```
1  function castVoteBySig(pID, support, v, r, s)  {...
2      bytes32 structHash = keccak256(BALLOT_TYPEHASH,
         pID, support);
3      bytes32 digest = keccak256("\x19\x01",
         domainSeparator, structHash);
4      address signer = ecrecover(digest, v, r, s);...}
```

**Listing 3: Compound DApp Back-end**

Upon further investigation of the smart contract side of *Compound* in Listing 3, SIGSCOPE ascertains that this DApp uses *eth_sign TypedData_v4* with a correct DS, but it fails to implement nonce and deadline, rendering it vulnerable to repetitive replay attack.

## 5.6 Ethical Best Practices

**Bug disclosure**: We identify the developers of vulnerable DApps detected by SIGSCOPE. Using Etherscan's ethmail service and official contact information (e.g., DApp websites, Discord, X, GitHub), we verify ownership and notify them of the vulnerabilities. The developers' responses are documented in the SIGSCOPE repository. **Use of data**: The code analyzed in this work is collected from public places, such as JavaScript from public websites and smart contracts crawled from Ethereum blockchains. To protect vulnerable DApps, we will mask the DApp names and other identifiable information in the camera-ready, such as the ones in the Case Study Section 5.5, so that they will not be attacked after paper publication.

## 6 Related Work

We extend related works in Appendix A.14 by categorizing 30 works into three areas of security analysis: Smart Contract Security Analysis, Front-end Security Analysis, and DApp Security Analysis. To the best of our knowledge, no research aids the community in comprehending and automatically identifying the emerging off-chain message signing-related vulnerabilities in DApps.

## 7 Discussion and Mitigation

One limitation of SIGSCOPE is its inability to analyze closed-source DApps and smart contracts, presenting a real-world constraint. However, since most impactful DApps are open-source to foster user trust, this limitation may be less significant. Additionally, SIGSCOPE currently supports only smart contracts written in Solidity and lacks support for languages like Vyper. The same applies to DApps' front-ends; SIGSCOPE can analyze those built with JavaScript, TypeScript, and HTML but cannot handle obfuscated JavaScript. Fortunately, over 97% of the top 100 DApps use these languages. To analyze a wider range of DApps, SIGSCOPE needs to support more languages.

## 8 Conclusion

In this work, our study delves into the security aspects of off-chain message signing methods, identifying various vulnerabilities related to signing and verification processes and illustrating various attack scenarios. We introduce SIGSCOPE, a novel automated static analysis framework capable of conducting hybrid code analysis on both the front-end and back-end of DApps. This framework allows for the detection and analysis of off-chain message signing methods and associated security vulnerabilities. Our evaluation of SIGSCOPE on a large dataset of 4937 DApps, including the top 100, demonstrates its effectiveness in identifying DApps with vulnerabilities related to off-chain message signing.

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

# A Appendix

## A.1 Blockchains, Smart Contracts, and DApps

Blockchain technology, introduced in 2008 [44], is a secure and decentralized ledger system that operates over a peer-to-peer network. This ledger comprises a series of blocks that resist modification and uphold the principle of non-repudiation. Each block holds a collection of transaction data and relevant state information, including the time of creation, transaction hash, preceding block, and more.

Originally, blockchain technology was primarily employed to record cryptocurrency transactions. Yet, it was not long before the exploration of additional applications began. Ethereum [27], notable as one of the leading mainstream blockchains, was the pioneer in allowing the creation of smart contracts, which are essentially programs written in high-level programming languages (e.g., Solidity, Vyper) and executed on blockchains. This innovation allows users to establish their own rules for ownership, transaction structures, and the state transition. Smart contracts are now a key driver in transforming various industries, such as NFT marketplaces [40, 48] and decentralized finance (DeFi) [19, 61].

A decentralized application, a.k.a. DApp, is structured with a front-end for user interaction and a back-end smart contract. The front-end not only provides the user interface but also includes JavaScript code for communication with other elements, such as the back-end and cryptocurrency wallets. The back-end smart contract, situated on a blockchain, receives user inputs through the front-end and processes function calls, which are effectively blockchain transactions. These function executions lead to modifications in the blockchain's stored data. Due to the inherent properties of blockchain, smart contracts are immutable post-deployment, assuring robust security since not even the developers can alter the code. This immutable nature has significantly contributed to the growth of DApps, with 2023 witnessing the development of 3,000 DApps and daily unique active wallets (UAW) of 4.2M [22].

## A.2 eth_signTypedData Example

```
1  gatherPermitSignature: async function
       gatherPermitSignature() {...
2    const message = ...
3
4    const domain = permitInfo.version
5    ... {name: permitInfo.name,
6        version: permitInfo.version,
7        verifyingContract: tokenAddress,
8        chainId,}...
9
10   const data = JSON.stringify({types: {...,
11       domain,
12       primaryType: 'Permit',
13       message,})
14
15   return provider.send('eth_signTypedData_v4', [
           account, data])...
```

Listing 4: UniswapV2 Front-end Message Signing

```
1  mapping(address => uint) public ns;
2  function permit(..., v, r, s) external {
3      require(deadline >= block.timestamp, 'UniswapV2:
           EXPIRED');
4      ...
5      bytes32 digest = keccak256(abi.encodePacked
6          ('\x19\x01',DOMAIN_SEPARATOR,keccak256
7              (abi.encode(..., value, ns[owner]++,
                   deadline))));
8
9      address recoveredAddress =
10         ecrecover(digest, v, r, s);
11
12     require(recoveredAddress != address(0) &&
13         recoveredAddress == owner,
14         'UniswapV2: INVALID_SIGNATURE');
15
16     _approve(owner, spender, value);}}
```

Listing 5: UniswapV2 Back-end Signature Verification

```
1  const NN = tokenNonceState.result?.[0]?.toNumber()
2    if (tokenNonceState.loading || typeof NN !== '
       number')
3      return {
4        state: UseERC20PermitState.LOADING,
5        signatureData: null,
6        gatherPermitSignature: null,}
7
8  const signatureDeadline = transactionDeadline.
       toNumber() + PERMIT_VALIDITY_BUFFER
9
10 const message = ...{
11       owner: account,
12       spender,
13       value,
14       nonce: NN,
15       deadline: signatureDeadline,}
```

**Listing 6: UniswapV2 Front-end Message Construction**

## A.3 Details of the Proposed Attacks

*A.3.1 Impersonation Attack.* This attack is associated with the missing secure prefix and directly exploits *eth_sign* with three steps. First, the attacker carefully constructs a tailored desired transaction (e.g., transfer funds from the victim user's wallet to the attacker's desired wallet) as shown in Listing 7. Note that to execute a transaction, the attacker only needs to sign the hash of the transaction content using the victim's private key. The attacker then hashes the transaction content using a specific hash function, normally *Keccak-256*. The generated hash value is ready to be signed by the victim user.

```
1  TX= {
2    from:0x3E0DeFb880cd8e163baD68ABe66437f99A7A8A74
3    to: 0xa2c0946aD444DCCf990394C5cBe019a858A945bD,
4    gas: "0x1312d00",
5    gasPrice: "0x12ff2ba729",
6    maxFeePerGas: "0x29bc1f157c",
7    nonce: "0x81",
8    input: "0x8a10f9ce00 ..., ...}
9  hash= "0xee038a31ab6e3f06bd747ab9dd0c3abafa48
10           a51e969bcb666ecd3f22ff989589"
```

**Listing 7: Real-world Attack Transferring NFTs [28]**

Second, the attacker aims to entice the user into starting a signature challenge and persuade them to sign the message, which is, in this case, a transaction hash. Upon acquiring the user's signature on the desired transaction hash, in the last step, the attacker can submit this signed transaction to a blockchain network and wait for the transaction to execute, which will trigger the unauthorized token transfer without any necessary on-chain action by the victim user.

This exploitation relies on the fact that a transaction's metadata comprises 66 hexadecimal with no secure prefix, including the '0x' prefix generated through *Keccak256* calculation, obscuring its precise meaning from users. Consequently, users may unwittingly consent to the transaction, effectively granting access to their pending authorization. An obscured signature request shown in Figure 5 illustrates this attack scheme, where the signer remains unaware of the exact message content they are validating.

*A.3.2 Replay Attack.* Replay attack exploits the improper or missing security features in the signing and verification processes to

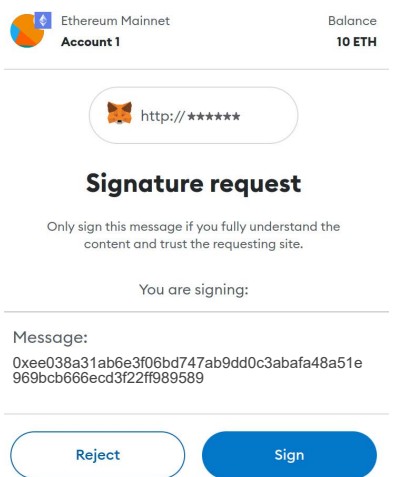

**Figure 5: Blind Signature Challenge: Listing 7 Hash**

abuse a signature and subsequently replay it with malicious intentions. For instance, if a user sends a signed message to a DApp that contains a contract M to perform the verification, an attacker can steal the signature from a legitimate DApp and replay the same signature to another contract N without proper authorization.

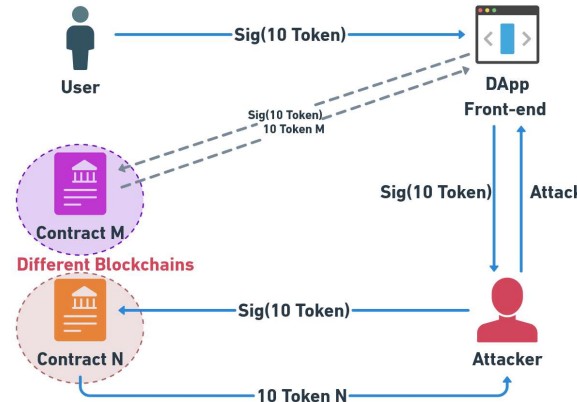

**Figure 6: Type 1 Replay Attack**

We break down the replay attack into four types based on different scenarios. Assuming the user signs a message authorizing a transfer of 10 tokens associated with contract M to a desired wallet, the attacker can perform an attack and obtain the signature, leading to four possible scenarios.

**Type 1: Cross-chain Replay.** The signature can be used to call contract M and withdraw 10 associated tokens from the user's wallet. However, if the signature does not contain a DS or has a DS containing an incorrect implementation of the chainId, the attacker may use that same signature to call another contract N instead of contract M. In this case, contract N can be located on a different blockchain. Consequently, the attacker could withdraw 10 tokens N from the user's wallet, as illustrated in Figure 6. Tokens N are different and could be more valuable than the tokens M.
**Type 2: Same-chain Replay.** The scenario for this type of Replay attack is the same as type 1 except that both contract M and contract N are located on the same blockchain, meaning that token M

**Figure 7: Type 2 Replay Attack**

and token N are native to the same blockchain as illustrated in Figure 7. When the signature is missing DS or contains an incorrectly implemented VC, it opens up a loophole to this type of attack.

**Type 3: Repetitive Replay.** In this scenario, the attacker possesses the signature and employs it to invoke contract M. However, if the signature is missing or has improper `nonce` and `deadline`, the attacker can leverage the same signature and make repetitive calls to contract M instead of just once. This repetitive invocation ultimately leads to the withdrawal of all tokens associated with contract M from the user's wallet. As depicted in Figure 8, if the user has 100 tokens linked to contract M and has only signed a message to transfer 10, the attacker can exploit the signature to call contract M ten times, withdrawing all 100 tokens from the user's wallet.

**Type 4: Complete Replay.** This scenario combines elements from the previous scenarios, allowing the attacker to invoke multiple contracts (e.g., both contracts M and N) repeatedly using the same signature. This situation arises when all three security features, namely DS (VC part), `nonce`, and `deadline`, are either missed or improperly implemented in the signature. The attacker can systematically withdraw all distinct tokens associated with those contracts from the user's wallet. This is more severe than the previous types, as it has the potential to rapidly deplete the user's wallet using a single signature. More explanation is provided in the Appendix A.11.

## A.4 System overview

SigScope analyzes both the front-end and back-end source code to identify security issues and aggregates them to generate comprehensive results showcasing DApps signing-related security issues. It produces analysis results pertaining to security inference and reporting. During the pre-processing step, SigScope determines whether the DApp utilizes off-chain signing and verification methods. If affirmative, the analysis core is activated to extract signing and verification security-related features, characterizing the signing and verification process. Subsequently, the security inference analyzer vets the DApp, uncovering potential security issues.

To this end, SigScope performs static intra- and inter-procedural context-sensitive, flow-insensitive analyses of the DApp atop state-of-the-art analysis frameworks CodeQL [18] and Slither [52]. CodeQL examines the front-end code of DApps, such as JavaScript, by representing the codebase as a structured database. It then employs

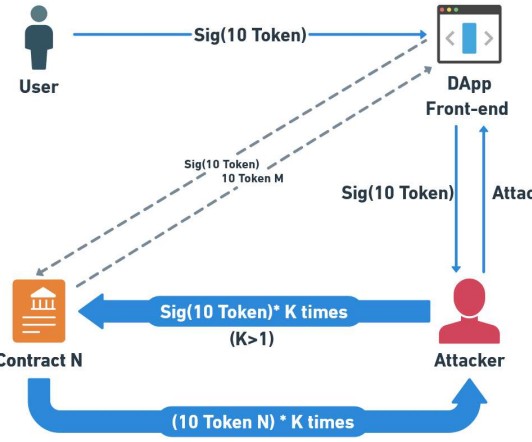

**Figure 8: Type 3 Replay Attack**

the CodeQL query language to interrogate this database, enabling developers to explore code properties and patterns. Slither is a Solidity smart contract static analysis framework that runs a suite of detectors and prints information about contract logic details and code analysis. Slither transforms the contract code into a static assignment intermediate representation (IR) named SlithIR, which facilitates implementation while preserving semantic information.

As depicted in Figure 2, SigScope functions as an analysis framework comprising an analysis core and four major tools. SigScope receives two inputs: the smart contract source code and the front-end source code, which includes JavaScript, Typescript, and HTML files.

## A.5 Pre-processing

To determine whether a DApp is entangled with off-chain message signing-related security vulnerabilities, SigScope initiates an analysis to identify the presence of message signing and verification methods. However, to ensure the effectiveness of the identification, it is essential to eliminate unreachable code regions, as a DApp's code may contain some instances of signing methods that can never be invoked. Unreachable code elimination thus ensures the effectiveness of SigScope's detection capability.

**Signing and Verification Method Identification.** The first task in the pre-processing phase is to identify all the signing and verification methods in a given DApp to facilitate further vulnerability detection. The easiest way is to scan over the whole code base in both the front-end and the back-end, searching for some API names (e.g., *personal_sign* and *ecrecover*). However, this approach will result in high false positives since it includes keywords in comments. To avoid this, SigScope generates control-flow graphs for both sides and iterates over all the basic blocks to search for external function calls to these APIs and consider the caller functions of APIs as the signing and verification methods in the DApp.

**Unreachable Code Elimination.** After identifying all the signing and verification methods, SigScope aims to eliminate the unreachable ones to guarantee the identified methods are indeed invokable. Traditionally, researchers have employed static inter-procedural constant propagation to identify unreachable code in C programs,

focusing on locating conditional branches that could never be executed [24]. However, this top-down strategy is not ideally suited for us for two reasons: 1) it is time-consuming since it aims to identify all unreachable code segments across the entire program, while our objective is specifically to isolate unreachable signing and verification methods; 2) unlike C programs, Solidity and JavaScript often feature multiple points of entry, complicating the application of this method. In response, we adopt a bottom-up methodology. Our system generates inter-procedural control-flow graphs (ICFGs) for both the front-end and back-end, extracts all the guarding conditions and the root caller of each signing and verification method (i.e., an entry point that eventually invokes the method) by searching backward from each identified signing and verification method, and performs backward dataflow analysis on the variables in the guarding conditions to examine whether the condition is satisfiable. We deem one method as reachable if it is not guarded by an unsatisfiable condition and its root caller exists. For the JavaScript code, we need to ensure that the root caller is further called by the HTML code.

## A.6    Back-end Code Analysis

After the pre-processing step, SigScope performs back-end code analysis on the smart contracts on top of Slither to extract verification-related information from the verification process of a DApp. This phase is responsible for analyzing three major security features: SVCheck, deadline, and nonce. Additionally, it infers the structure of the signed message, which is vital for front-end analysis, specifically identifying the positions of nonce and deadline in the signed message. This is why we perform the back-end analysis first.

while Slither handle basic analysis on a static assignment intermediate representation (IR) named SlithIR, it does not specifically address the identification of nonce and deadline from a message and accurately recording their positions. Therefore, we need to implement custom callgraph, forward and backward dataflow analysis techniques.

A simple approach is to create a keyword list for each element and rely on program symbols, such as function and variable names, to identify them. For instance, if a variable in a verification method is named "deadline", it is likely to be the deadline in the message. However, this approach can lead to high inaccuracy since the keyword list can never be exhaustive. Instead, we rely on the intrinsic characteristics of these security features to identify them. Algorithm 1 outlines our comprehensive approach for conducting back-end verification analysis. The algorithm is designed to accept three key inputs: the back-end smart contract ($sc$), the verification function identified during pre-processing ($vf$), and the root caller function of the verification process ($rc$).

To this end, SigScope begins by generating the call graph of $sc$ and producing $F_{cp}$, which is a set of functions that are covered in the call path from $rc$ to $vf$ in $sc$ Then, the algorithm detects the existence of $SVCheck$ (Ln.3-9). It first identifies the return value of an $ecrecover$ function $ret_{ecrecover}$ and performs forward dataflow analysis on it to create a set of statements $\mathbb{SINK}$ (Ln.3-4). If a conditional check $sink$ exists in the set, we analyze the condition of $sink$ to extract the variable that is compared against $ret_{ecrecover}$ (Ln.5-6). We further perform backward dataflow analysis on the variable

to extract its data origin. If the data originated from a parameter $rc$, which means the recovered signer of the message is compared against part of the signed message (the original signer), then we believe $SVCheck$ is enforced (Ln.7-9).

To achieve this, SigScope begins by generating the call graph of $sc$ and deriving $F_{cp}$, which encompasses all functions traversed in the call path from $rc$ to $vf$ within $sc$. This step ensures that the analysis focuses specifically on the relevant portions of the smart contract where verification operations occur. Subsequently, the algorithm proceeds to verify the presence of $SVCheck$ (Ln. 3-9). It begins by capturing the return value $ret_{ecrecover}$ from an $ecrecover$ function call and conducts forward dataflow analysis to compile a set of statements $\mathbb{SINK}$ (Ln.3-4). This set includes potential locations in the code where the value $ret_{ecrecover}$ might be used in conditional checks or comparisons. If a conditional check $sink$ is identified within $\mathbb{SINK}$, the algorithm further investigates the condition associated with $sink$ to identify the variable being compared against $ret_{ecrecover}$ (Ln. 5-6). This involves analyzing the control flow to understand the context in which the comparison occurs and the significance of the comparison result. To validate $SVCheck$, the algorithm performs backward dataflow analysis on the identified variable to trace its origin. If the variable can be traced back to a parameter $rc$, indicating that the recovered signer of the message corresponds to the original signer as intended by the DApp's logic, then $SVCheck$ is inferred to be enforced (Ln. 7-9). This verification process ensures that the smart contract code effectively validates the authenticity and integrity of messages signed off-chain. By meticulously analyzing the flow of data and control within the smart contract, SigScope provides a robust assessment of $SVCheck$ implementation, enhancing the security and reliability of DApps utilizing such verification mechanisms.

Subsequently, the algorithm proceeds to analyze the $Deadline$ and $Nonce$ parameters in a comprehensive manner (Ln.10-20). It begins by identifying all the arguments ($\mathbb{ARGS}$) of the hash function $Keccak$ and then conducts a backward dataflow analysis on each argument $arg$ to generate a set $\mathbb{SRC}$ (Ln.10-12). This step is crucial as it traces the origins and dependencies of each argument, providing insight into their roles and interactions within the function. If a conditional check $src$ is found within the set that compares $arg$ against the current time indicators (such as $Blocknumber$ or $Timestamp$), the algorithm infers that a $Deadline$ parameter is present. It then records the position of this '$Deadline$' parameter, which corresponds to the position of $arg$ within the parameter list (Ln.13-17). This identification process is vital for understanding time-sensitive operations within the contract.

Furthermore, if another $src$ in $\mathbb{SRC}$ is determined to be data-dependent on a specific parameter $rc$ (where the nonce is unique to each signer), and this parameter is incremented with each verification while its value originates from a non-local mapping variable, the algorithm deduces that a $Nonce$ parameter is present. The position of the $Nonce$ is then documented. Additionally, the algorithm may consider other contextual clues and cross-references within the code to strengthen its deductions about the $Deadline$ and $Nonce$. By examining usage patterns, dependencies, and the broader logical flow of the program, the algorithm enhances the accuracy and reliability of its analysis. Finally, the algorithm compiles and returns a comprehensive report detailing the existence and precise positions

of the three security features: the *SVCheck*, the *Nonce*, and the *Deadline*.

## A.7 Front-end Code Analysis

After thoroughly analyzing the verification process within the back-end smart contract, SɪɢScᴏᴘᴇ must extract and verify the proper implementation of security-related features in the front-end, specifically the *DS*, *Deadline*, and *Nonce* parameters. This ensures that the entire system maintains the integrity and security of these critical components across both the front-end and back-end environments.

Although CodeQL handles basic dataflow analysis by representing the codebase as a structured database and using the CodeQL query language to interrogate this database, it does not specifically address extracting and verifying the proper implementation of security-related features. Therefore, we must implement custom backward data flow analysis techniques as follow.

SɪɢScᴏᴘᴇ employs algorithm 2, designed to handle this aspect of the verification process. The algorithm takes as inputs the JavaScript code (*js*), the identified signing methods from pre-processing (*sm*), and the positions of the *Deadline* and *Nonce* from the previous phase ($POS_{ddl}$ and $POS_{nonce}$), and it outputs the analysis results for *DS*, *Deadline*, and *Nonce*. SɪɢScᴏᴘᴇ also analyzes to detect the existence of Security Prefix by utilizing pre-processing information about the type of signing methods (*sm*) used. It further conducts analysis in the front-end to identify patterns such as "\x19Ethereum Signed Message:\n<length of message>" by parsing the message body. This ensures the detection and proper handling of security prefixes within the signing process.

The algorithm begins by locating the message data section variable $V_{data}$ by examining the corresponding parameter in the signing method *sm* (e.g., the second parameter in *signTypedData_v*4) (Ln.1). It then performs backward dataflow analysis on $V_{data}$ to identify the message section (*message*) and the domain separator ('*DS*'), as $V_{data}$ is data-dependent on both (Ln.2-4). Subsequently, it analyzes *DS* to ensure it enforces a valid `chainID` and `verifyingContract` (Ln.5).

The algorithm then focuses on the *Deadline* parameter (Ln.6-10). Utilizing $POS_{ddl}$ from the back-end, it identifies the variable that holds the deadline information ($V_{ddl}$). Backward dataflow analysis is conducted on '$V_{ddl}$' to verify that the value is derived from the current time plus a predefined small period. The analysis establishes *Deadline* and its order position within the signature, based on its data dependency on time (e.g., *block.number* or *block.timestamp*), with its source coming from a function input argument. This confirms whether the '*Deadline*' is correctly implemented (Ln.7-10).

A similar approach is taken for the '*Nonce*' parameter (Ln.11-15). The algorithm checks whether the '*Nonce*' value originates from a call to the back-end, ensuring that it is correctly implemented. This involves verifying that the '*Nonce*' is incremented appropriately and is signer-specific which means considering its data dependency on the signer's address.

To further extend the verification process, the algorithm might include cross-referencing additional contextual clues within the front-end code. This includes examining usage patterns, dependencies, and the overall logical flow of the program. Such comprehensive analysis strengthens the reliability of the deductions about '*DS*',

'*Deadline*', and '*Nonce*'. Eventually, the algorithm compiles and returns a detailed report on the analysis results for '*DS*', '*Deadline*', '*Nonce*', and Secure Prefix. This report includes the value, positions, and verification status of these parameters, providing a clear understanding of their implementation and ensuring that the front-end code aligns with the security requirements established in the back-end smart contract. This robust approach verifies the integrity and correctness of these security features. By meticulously validating both the front-end and back-end implementations, SɪɢScᴏᴘᴇ ensures a comprehensive and reliable security verification process, safeguarding the system against potential vulnerabilities and ensuring the integrity of critical security parameters.

## A.8 False Positives in Effectiveness

Furthermore, we investigate the 5 false positive cases to understand their root causes. Three of these cases involve scenarios where verification procedures are defined within external inherited smart contracts (e.g., external libraries), which increases the complexity of the analysis. Of these, two are mislabeled as vulnerable due to incorrect implementations of `nonce`, while one is misclassified due to errors in both `nonce` and `deadline` implementations. SɪɢScᴏᴘᴇ fails to capture the complete implementation details and security features in these cases, leading to their mislabeling as having reported vulnerabilities. The remaining two cases are caused by JavaScript code complexity, which leads to a failure in the backward dataflow analysis of `nonce`. This failure prevents SɪɢScᴏᴘᴇ from determining whether the nonce value originates from a back-end call, thus hindering its ability to assess the correctness of the implementation.

## A.9 Efficiency

We utilize the 159 verified DApps containing off-chain message signing to assess the runtime performance of SɪɢScᴏᴘᴇ. On average, SɪɢScᴏᴘᴇ takes 15.63 seconds and 42.74 seconds to perform the back-end and front-end analyses, respectively. As illustrated in Figure 9, SɪɢScᴏᴘᴇ can process more than 82.3% (131 out of 159) of DApps in under one minute, with a maximum processing time of 321.2 seconds. Consequently, we can conclude that the efficiency of SɪɢScᴏᴘᴇ is high enough for performing large-scale DApps analysis.

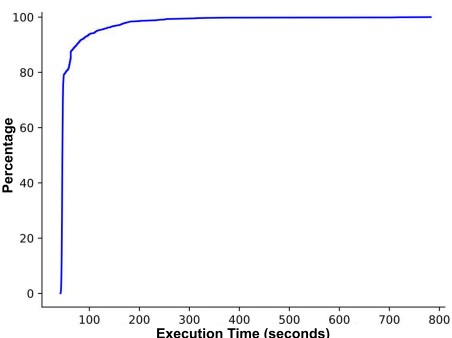

**Figure 9: CDF Diagram of SɪɢScᴏᴘᴇ Execution Time**

## A.10 Growth Trend of DApps

We also analyze the creation dates of the 1,579 detected DApps to explore the growth trend of DApps utilizing off-chain message signing. Figure 10 illustrates the increasing popularity of these

DApps over recent years, with a notable surge in deployments around 2022. This trend highlights the rapid adoption of off-chain message signing in the blockchain ecosystem.

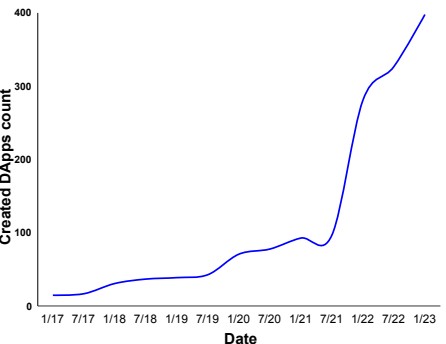

**Figure 10: DApps Using Off-chain Signing Growth Trend**

## A.11 Vulnerabilities & Attack Senarios

**Missing Secure Prefix.** As detailed earlier in Section 3, it leads to blind signature results, where the signer remains unaware of the exact message content they are signing. This vulnerability exposes the workflow to impersonation, enabling attackers to fabricate and deploy transactions that can transfer arbitrary amounts of funds from the victim's wallet.

**Missing or Improper DS.** If a DApp has a DS containing an incorrect implementation of the chainId, it is vulnerable to Cross-chain Replay attacks. If it contains a DS by an incorrectly implemented VC, it is vulnerable to Same-chain Replay attacks. If the signature does not contain a DS, it is vulnerable to both Cross-chain Replay and Same-chain Replay attacks. These replay scenarios enable an attacker to reuse a signature, initially intended for one specific DApp, to exploit the victim's funds through other unauthorized DApps at least once per DApp, regardless of whether nonce or deadline is correctly implemented.

**Missing or Improper Nonce.** This vulnerability leads to a Repetitive Replay, where the attacker can leverage the same signature to make repetitive calls to the contract instead of just once, allowing them to withdraw all tokens associated with the contract from the user's wallet.

**Missing or Improper Deadline.** By working in conjunction with nonce, deadline can mitigate the risk of Repetitive Replay by setting an expiration date for a signature. However, in Complete Replay, where an attacker can invoke multiple contracts repeatedly, the role of deadline is crucial. Even correctly implemented nonce cannot mitigate this scenario without the proper implementation of deadline.

**Missing or Improper Validity Check.** Regardless of the specific attack scenario, this vulnerability could allow an attacker to exploit it for profit, resulting in unauthorized changes to the state of the blockchain. This undermines the security of the entire Signer Verification phase.

## A.12 False Positives

Additionally, we conduct a detailed examination of the 49 false positive cases to uncover their underlying causes.

Out of these cases, seventeen involve situations where verification procedures are implemented in external inherited smart contracts, such as external libraries. This setup introduces additional complexity to the analysis process. Among these, nine cases are misclassified as vulnerable due to incorrect implementations of the nonce, while seven cases are incorrectly identified as vulnerable due to issues with the deadline implementations. In these instances, SigScope fails to accurately capture the complete implementation details and security features present in the external contracts, leading to erroneous conclusions about the existence of vulnerabilities.

The remaining 32 cases are due to front-end code complexity, such as JavaScript, which causes failures when performing backward dataflow analysis on $V_{data}$, as detailed in Algorithm 2. This analysis aims to identify the message section, *message*, and the domain separator $DS$ but fails because $V_{data}$ is data-dependent on both of these elements (Lines 2-4). In six cases, SigScope is unable to find the implementation details of $DS$ to determine whether it enforces a valid chainID and verifyingContract.

SigScope utilizes $POS_{ddl}$ from the back-end to locate the variable containing the deadline information ($V_{ddl}$) and performs backward dataflow analysis on it to check whether the value results from the current time plus a certain small period. In eleven cases, SigScope's *deadline* cannot correctly verify deadline due to failures in this backward dataflow analysis.

SigScope conducts a similar analysis on nonce to determine if its value originates from a call to the back-end and assess the correctness of nonce. Failures in the backward dataflow analysis of nonce prevent SigScope from determining whether the nonce value comes from a back-end call, thereby hindering its ability to evaluate the correctness of the implementation in sixteen cases.

## A.13 Case Study Detail

```
1   const deadline = 99999999999999;
2   const signature = await signer._signTypedData(
3   {     version: "2",
4       name: "USD Coin",
5       chainId: chainID,
6
7       verifyingContract: usdcTokenAddress,},...
8   {     owner: account,
9       spender: teslaContract,
10      value: ethers.utils.parseUnits(usdc.toString(),
            usdcDecimals),
11      nonce: (await usdcContract?.nonces(account))||0,
            deadline, });
```

**Listing 8: Synthetix Front-end**

**Case 2: Incorrect implementation of security measures.** For the second case study, we investigate Synthetix [56], a popular DeFi ecosystem with a $668M+ TVL. Based on SigScope analysis, this DApp utilizes *eth_signTypedData_v4* with a DS, nonce, and deadline; however, it does not implement deadline properly to prevent replay attacks. A proper implementation of deadline in the front-end should always be calculated as the current time (e.g., *timestamp*) plus a short time frame to eliminate the possibility of reusing the signature after a short period. However, as shown in Listing 8, the dealine value is assigned a large constant value

(Ln.1), making the DApp vulnerable to attacks because a signature can now be used indefinitely.

```
1  function permit(...) external {
2      if (deadline < block.timestamp)
3          revert();
4      address recoveredAddress = ecrecover(
5          keccak256("\x19\x01",DOMAIN_SEPARATOR(),
6              keccak256(
7                  keccak256("Permit(...)"),
8                      owner, spender, value,
9                      nonce, deadline)))), v,r,s);
10     store.nonces[owner] = nonce + 1;
11     if (recoveredAddress == address(0) ||
           recoveredAddress != owner) revert ();
12     _approve(recoveredAddress, spender, value);}
```

**Listing 9: Synthetix Back-end**

## A.14    Related-work Details

**Smart Contract Security Analysis.** Some research employs static analysis to enhance smart contract security and efficiency. USCHUNT [15] explores the balance between adaptability and security in upgradeable contracts. Madmax [32] targets vulnerabilities to prevent execution failures, while Slither [52] and Smartcheck [57] automatically detect flaws in Solidity contracts. Symbolic execution is also used to improve security; Mythril [20] analyzes EVM bytecode, EthBMC [31] combines symbolic execution with concrete validation, and Reguard [39] and Manticore [47] identify reentrancy and other bugs. Smartian [17] integrates fuzzing with static and dynamic analysis, while Confuzzius [30] leverages data dependency insights for fuzzing. CRYSOL [70] applies fuzzing to detect cryptographic defects in contracts. ContractFuzzer [34] and Sfuzz [45] apply fuzzing to uncover security issues.

Research highlights various formal verification methods to enhance smart contract security. Sailfish [16] improves state inconsistency detection, while VetSC.[25] extends DApp verification. Zeus[35] and Verx [49] focus on contract safety and condition verification. Smartpulse [54] analyzes time-based properties, Securify [60] identifies security breaches, and Verismart [53] ensures contract safety.

**Front-end Security Analysis.** These studies advance JavaScript security analysis using static and hybrid methods. Tripp et al.[59] enhance web security by combining static and dynamic analysis. Sun and Ryu[55] survey JavaScript analysis challenges. JSAI [38] targets dynamic aspects for vulnerability detection, while GATEKEEPER [33] enforces static security policies. Fang et al.[29] use semantic analysis to detect malicious code, and FAST[36] applies abstract interpretation for Node.js vulnerabilities. CoCo [69] focuses on browser extension security, and PROBETHEPROTO [37] addresses prototype pollution vulnerabilities.

**DApp Security Analysis.** DAppScan [72] creates large-scale datasets for detecting weaknesses in DApps' contracts and introduces the Smart Contract Weakness Classification Registry, including issues like Signature Malleability and Missing Protection against Replay Attacks. Ye et al.[68] propose methods for detecting DApp vulnerabilities. Darcher[71] addresses challenges in synchronizing on-chain and off-chain data in Ethereum-based DApps. Xue et al. [66] focus on reentrancy vulnerabilities.

