# OpenReview forum: "SigScope: Detecting and Understanding Off-Chain Message Signing-related Vulnerabilities in Decentralized Applications"
_ACM.org/TheWebConf/2025/Conference — WWW 2025 Poster_

### Official Review · Reviewer_F6rr · 2024-11-25

**Novelty:** 1
**Technical Quality:** 1

**Review:**

The paper claims to be the first systematic study on off-chain message signing vulnerabilities. However, the proposed SigScope framework focuses solely on a specific type of vulnerability—off-chain message signing. While such vulnerabilities do exist, their scope is relatively narrow and fail to significantly enhance the capabilities of existing smart contract and DApp security analysis tools. Notably, the paper lacks a comprehensive baseline comparison with existing security analysis tools, making it difficult to quantify SigScope actual effectiveness. In my opinion, it should be rejected.

**Questions:**

1.	Although the paper reports a high F1 score for SigScope on 494 samples, it lacks a more detailed performance analysis, including metrics such as recall, accuracy, and precision, making it difficult to effectively validate the tool's overall performance. The paper does not provide an in-depth discussion of potential false negatives, focusing only on the false positive rate (4.2%). This might obscure the tool's actual detection capabilities.
2.	The proposed SigScope framework involves a combination of static analysis for both the front-end (HTML/JavaScript) and back-end (smart contracts), but it does not provide performance and scalability metrics, such as runtime and memory consumption, when analyzing complex DApps. This makes it challenging to assess its efficiency in real-world applications.

**Reviewer Confidence:**

4: The reviewer is certain that the evaluation is correct and very familiar with the relevant literature

**Scope:**

3: The work is somewhat relevant to the Web and to the track, and is of narrow interest to a sub-community

---

### Official Review · Reviewer_hHDW · 2024-12-02

**Novelty:** 2
**Technical Quality:** 2

**Review:**

The paper presents SigScope, a novel static analysis framework designed to detect and understand security vulnerabilities related to off-chain message signing in decentralized applications (DApps).

Pros:
1. The hybrid analysis technique and the holistic assessment of DApps' security are innovative contributions to the field of blockchain security.
2. The findings have practical implications for the security of decentralized applications, highlighting common vulnerabilities and providing a tool to detect them.
3. The artifact of this paper is made publicly available.

Cons:
1. The challenges of implementing well-established static analysis in a hybrid way are not well elaborated, which raises concerns regarding the novelty and contribution of this work. According to Section 4, back-end smart contracts are analyzed before it was deployed on the blockchain, it is concerned whether it could be called as “hybrid”. While the design of SigScope is comprehensive, its core algorithms largely rely on the combination of existing technologies (e.g. back-end code on top of Slither), rather than introducing a novel analytical framework. To be more specific, the system depends on well-defined signature method standards and known vulnerability patterns, what would be the difficulties of building an automated detection process?
2. The motivation of this work is not well elaborated. First, the off-chain message signing defined by [51] in Section 1 lacks authoritative depth, which hinders establishing academic understanding of the core problem authors are trying to solve. Second, Section 3.5 (Proposed Attacks) is oversimplified in the main body, while [19, 56, 50] are not pointed to any academic paper or a real-world smart contract address as expected. The authors are supposed to list out real-world examples and detailed evidence on the proposed attacks rather than saying “we have successfully implemented” in one sentence.
3. The evaluation of this work is insufficient. Only the accuracy of automatically detected vulnerabilities compared to manually identified vulnerabilities are evaluated in Section 5.3 and Section 5.4, but the authors do not discuss how SigScope outperforms existing tools such as CodeQL or Slither in terms of performance (scanning time).
4. SigScope currently does not support various programming languages like Vyper and obfuscated JavaScript, which may limit its applicability to DApps implemented in different ways.
5. The organization of the paper needs to be refined. Some important justifications are missed in the main body (e.g. Section 3.5), while many known aspects occupy most of the contents. What’s more, there are several typos and grammatical errors throughout the paper, which detract from its overall quality. E.g. “We also describe describe” in Section 2, “Listing 1: Carbon DApp Domain Seperator” in Section 3.3.

**Questions:**

1. The paper focuses primarily on Ethereum-based DApps. How generalizable are the findings to DApps on other blockchain platforms?
2. Why do we need to employ static analysis to the front-end code if they are finally sent to a consensus based blockchain network? Why can’t we just let the unlawful transactions be reverted by consensus protocol?
3. Blind signature is an open problem in blockchain security, why would scanning the front-end code help to mitigate the problem?

**Reviewer Confidence:**

4: The reviewer is certain that the evaluation is correct and very familiar with the relevant literature

**Scope:**

4: The work is relevant to the Web and to the track, and is of broad interest to the community

---

### Official Review · Reviewer_gKc8 · 2024-12-02

**Novelty:** 5
**Technical Quality:** 5

**Review:**

** Strength **

- Introduce a novel framework, SigScope, for identifying vulnerabilities in off-chain message signing within DApps.
- Define signing-related and verification-related vulnerabilities, accompanied by illustrative attack scenarios.
- Uncover 1,154 real-world vulnerabilities.

** Weakness **

- Missing false negative analysis.
- Unclear details on the outcomes of bug disclosures.

** Comments **

I appreciate the authors for proposing the first framework targeting vulnerabilities in the message signing and verification processes. I am also impressed by the discovery of such a significant number of vulnerabilities in real-world DApps. However, I believe that some critical information and evaluation are not included in the paper. The following enlists my concerns.

** Missing false negative analysis**
SigScope identified 1,203 vulnerabilities across 1,579 DApps based on the definitions provided in the paper. While the authors reported 49 cases where the identified vulnerabilities were false positives, they did not address the potential for false negatives. For instance, based on my understanding, SigScope detects the presence of the SVCheck but does not validate whether the check itself is implemented correctly. If the recovered signer or the original signer is incorrectly processed before reaching the conditional check, this could result in the same type of vulnerability going undetected. Since the authors did not evaluate whether SigScope might fail to identify such cases, I recommend compiling a set of known vulnerabilities in DApps and conducting an evaluation to verify whether SigScope can reliably detect them. If such an evaluation is infeasible, I suggest that the authors at least discuss the potential for false negatives in the paper.

** Missing Bug Disclosure Outcomes **
Section 5.6 mentions that the authors reported identified vulnerabilities to the corresponding vendors. Given that this paper introduces new attack scenarios, I am curious whether these scenarios have been acknowledged by real-world DApp vendors. I recommend that the authors include the status of the reported vulnerabilities in the paper.

** Missing information in the main body **
I noticed that a significant amount of crucial information and discussion is placed in the Appendix, while less critical content occupies the main body of the paper. Specifically, the possible attack scenarios (Sections A.3 and A.11), design details (Sections A.5–A.7), discussions on false positives (Sections A.8 and A.12), and related work (Section A.14) are essential for understanding the presented work. For example, without a clear understanding of the attack scenarios, the security implications of the identified vulnerabilities remain ambiguous. I suggest that the authors consider making Sections 1 and 2.2 more concise to incorporate this important information into the main body. Additionally, to create space, I recommend summarizing the results of Figures 3 and 4 in textual form instead of presenting them as graphs.

**Questions:**

- Is this the first work to identify the vulnerabilities discussed in Sections 3.3 and 3.4?
- Did the authors report all identified vulnerabilities to the vendors? If so, how many of these vulnerabilities have been acknowledged or patched?
- What are the potential false negative cases that SigScope may exhibit?

**Reviewer Confidence:**

2: The reviewer is willing to defend the evaluation, but it is likely that the reviewer did not understand parts of the paper

**Scope:**

3: The work is somewhat relevant to the Web and to the track, and is of narrow interest to a sub-community

---

### Official Review · Reviewer_qZnf · 2024-12-02

**Novelty:** 3
**Technical Quality:** 6

**Review:**

The research is based on an evolving and significant subject inside decentralized applications (DApps), emphasizing off-chain message signing vulnerabilities. It follows a thorough process that includes static analysis tools and systematic evaluation. The dataset size is small but has strength in its diversity, strengthening the validity of the findings. The content of the paper is dense

SigScope is a novel hybrid approach that bridges frontend and backend vulnerability assessment in DApps. However, some foundational techniques, such as static code analysis, are not entirely new, which reduces the perception of originality in the methodology.

**Questions:**

1. Are there specific algorithms or techniques introduced in this work that are new to the field of static analysis?
2. Beyond identifying vulnerabilities, does SigScope provide actionable recommendations for mitigation?
3. Are there best practices that developers should adopt to prevent these vulnerabilities from occurring during the development phase?
4. Could SigScope itself be misused by attackers to identify vulnerable DApps? If so, what safeguards are in place to prevent this?

**Reviewer Confidence:**

2: The reviewer is willing to defend the evaluation, but it is likely that the reviewer did not understand parts of the paper

**Scope:**

3: The work is somewhat relevant to the Web and to the track, and is of narrow interest to a sub-community

---

### Official Review · Reviewer_o5P4 · 2024-12-02

**Novelty:** 7
**Technical Quality:** 6

**Review:**

The paper, SigScope, investigates the security of off-chain message signing in decentralized applications (DApps). Off-chain message signing is a technique used to improve the performance, cost-efficiency, and usability of DApps by allowing users to sign messages off-chain rather than signing transactions directly on the blockchain. However, it introduces new security risks as developers must design application-specific signatures, which is error-prone.

The authors analyze different signing methods and identify several security vulnerabilities that can arise. These vulnerabilities can be categorized into signing-related vulnerabilities and verification-related vulnerabilities.

To detect these vulnerabilities, the authors develop SigScope, a static analysis framework that analyzes both the front-end and back-end code of DApps. SigScope first identifies all instances of signing and verification methods and then analyzes the code to determine if the necessary security features are correctly implemented. The authors evaluate SigScope on a dataset of 4,937 real-world DApps and find that a significant number of DApps (1,154 or 73%) contain vulnerabilities related to off-chain message signing.

The paper presents a significant contribution to understanding and addressing off-chain message signing vulnerabilities in DApps.

## Pros
- This is the first systematic study to uncover and characterize security issues in off-chain message signing in DApps.
- The study considers both the front-end and back-end components of DApps, providing a comprehensive view of the security implications of off-chain message signing.
- The authors develop a practical tool, SigScope, which can be used to automatically detect vulnerabilities in real-world DApps.
Analysis of 4,937 DApps reveals a high prevalence of vulnerabilities, highlighting the importance of the study.
- The study includes case studies of vulnerabilities in popular DApps, showcasing the practical impact of the findings.

---

## Cons
- SigScope supports only Solidity for smart contracts and JavaScript, TypeScript, and HTML for front-end code, restricting its applicability to other languages and frameworks.
- The study identifies vulnerabilities but does not offer concrete solutions for developers to mitigate these risks.
- The paper includes a 7-page appendix with essential content such as code analysis, case studies, and discussions about false positives. While these details are valuable, their exclusion from the main paper raises concerns about whether the current 8-page limit of the venue is suitable for effectively presenting the research.

---

## Suggestions
- **Expand Language Support:** Include other smart contract languages (e.g., Vyper) and front-end frameworks to broaden applicability.
- **Enhance External Contract Analysis:** Address challenges with analyzing inherited smart contracts and propose techniques like dependency analysis to improve accuracy.
- **Include Mitigation Strategies:** Dedicate a section to practical solutions.
- **Validate False Negatives:** Strengthen the claim of 0% false-negative rate with additional methodologies.

---

## Spelling and Grammar
In section 2.1, the sentence "We also describe describe how off-chain message signing works..." repeats the word "describe."

**Questions:**

- Authors mentioned reporting the DApp vulnerabilities to the vendors. What responses have you received from vendors?

- Could the authors suggest potential mitigation techniques that developers could implement to address the various signing-related and verification-related vulnerabilities discussed in the paper?

- While the evaluation focuses on the false positive rate, the authors mention a 0% false-negative rate. Could the authors discuss the potential implications of undetected vulnerabilities (false negatives)? How might false negatives impact the overall security assessment of DApps?

- The authors relate 32 false positives to the complexity of JavaScript code, particularly in the backward dataflow analysis of Vdata. Could the authors expand on the types of JavaScript code complexities encountered during the analysis?

**Reviewer Confidence:**

3: The reviewer is confident but not certain that the evaluation is correct

**Scope:**

4: The work is relevant to the Web and to the track, and is of broad interest to the community